# Assessment of Effects of Laser Light Combining Three Wavelengths (450, 520 and 640 nm) on Temperature Increase and Depth of Tissue Lesions in an Ex Vivo Study

**DOI:** 10.3390/ma13235340

**Published:** 2020-11-25

**Authors:** Kamil Jurczyszyn, Witold Trzeciakowski, Zdzisław Woźniak, Piotr Ziółkowski, Mateusz Trafalski

**Affiliations:** 1Department of Dental Surgery, Wroclaw Medical University, 50-425 Wroclaw, Poland; mateusz.trafalski@umed.wroc.pl; 2Institute of High Pressure Physics, Polish Academy of Sciences, 01-142 Warsaw, Poland; wt@unipress.waw.pl; 3Department of Pathology, Wroclaw Medical University, 50-368 Wroclaw, Poland; zdzislaw.wozniak@umed.wroc.pl (Z.W.); piotr.ziolkowski@umed.wroc.pl (P.Z.)

**Keywords:** diode laser, multi-wavelength laser, diode laser surgery, interaction light-tissue

## Abstract

Background: Lasers are widely used in medicine in soft and hard tissue surgeries and biostimulation. Studies found in literature typically compare the effects of single-wavelength lasers on tissues or cell cultures. In our study, we used a diode laser capable of emitting three components of visible light (640 nm, red; 520 nm, green; 450 nm, blue) and combining them in a single beam. The aim of the study was to assess the effects of laser radiation in the visible spectrum on tissue in vitro, depending on the wavelength and pulse width. Methods: All irradiations were performed using the same output power (1.5 W). We used various duty cycles: 10, 50, 80 and 100% with 100 Hz frequency. Maximum superficial temperature, rate of temperature increase and lesion depth were investigated. Results: Maximum superficial temperature was observed for 450 + 520 nm irradiation (100% duty cycle). The highest rate of increase of temperature was noted for 450 + 520 nm (100% duty cycle). Maximum lesion depth was observed in case of three-wavelength irradiation (450 + 520 + 640 nm) for 100, 80 and 50% duty cycles. Conclusions: The synergistic effect of two-wavelength (450 + 520 nm) irradiation was observed in case of maximum temperature measurement. The deepest depth of lesion was noted after three-wavelength irradiation (450 + 520 + 640 nm).

## 1. Introduction

Lasers, including semiconductor lasers are widely used in medicine. They are mainly used in soft tissue surgery for cutting and cauterization [1,2]. They are also used in biostimulation and photodynamic therapy [3,4,5]. Their usefulness stems from their small size, low cost of operation and the use of flexible optical fibres, which have a positive impact on work comfort. Properties which characterise laser light include approximate monochromaticity, coherence and beam parallelism [6,7]. Laser beams can selectively affect chromophores found in tissues, as they absorb electromagnetic waves with predilection to their specific length [6]. In the visible range (400–700 nm), the absorption in the tissue is dominated by hemoglobin (with peaks in the blue and yellow region) and by melanin (decreasing monotonically towards longer wavelengths) [5,8]. The above properties are used to select the right laser device for given clinical needs. For the treatment of highly pigmented lesions and vascular lesions, light emitted by diode lasers (400–980 nm), pulsed dye lasers (580–595 nm), neodymium-doped yttrium aluminum garnet lasers (Nd:YAG, Nd:Y_3_Al_5_O_12_, 1064 nm) and frequency-doubled (using potassium titanyl phosphate, KTP) YAG lasers (532 nm) is most useful [9]. Carbon dioxide-based lasers (CO_2_, 10,600 nm) are very well suited to cutting soft tissue with high water content. A wavelength of 2940 nm, characteristic for erbium-doped YAG lasers (Er:YAG), is in turn strongly absorbed by water and hydroxyapatite. Due to this, Er:YAG lasers are used for both soft tissue cutting and bone preparation [1,10]. Apart from wavelength, tissue response is affected by power, energy density and radiation dose. An increase in power and energy density leads to intensification of thermal and ablative processes [11,12].

Depending on the value of power expressed as a function of time, lasers can be divided into continuous wave (CW) lasers and pulsed wave lasers. Operating parameters of a laser device can be further modified via pulse width and frequency. CW lasers provide a constant light beam, and the supplied energy is constant within a unit of time. Pulsed operation, with a constant pulse width, supplies a dose of energy, which is followed by an interval with no emission. Impulses repeat regularly, and their length over time (pulse width) is lower than the length of the period. Equality between pulse width and the length of the period indicates a CW laser. Pulsed operation is described by two parameters: frequency and duty cycle, that is, the ratio of pulse width to period, expressed as a percentage (100% for continuous wave lasers) [9]. Most laser systems emit pulses with a width between 0.1 ms and 100 ms. Even shorter pulse widths, in the order of a few nanoseconds (1.0 ns = 10^−9^ s), are characteristic for a mode of operation called Q-switch [13,14]. This mode is used mainly in removal of pigmented lesions, which include freckles, café au lait spots, lentigines and tattoos [15].

Studies found in literature typically compare the effects of single-wavelength lasers on tissues or cell cultures. In our study, we used a diode laser capable of emitting three components of visible light (RGB: 640 nm, red; 520 nm, green; 450 nm, blue) and combining them in a single beam. The aim of the study was to assess the effects of laser radiation in the visible spectrum on tissue in vitro, depending on wavelength and pulse width. This assessment concerned not only the individual RGB components, but also their combination during exposure. The effects on tissue were studied from physical (assessment of temperature using a thermal imaging camera) and histopathological points of view, the latter of which involved an assessment of the depth of lesions caused by irradiation. We found unexpected synergies while combining two different wavelengths simultaneously.

## 2. Materials and Methods 

### 2.1. Laser Irradiation

Irradiation was performed using a three-wavelength laser source (450 nm, 520 nm and 640 nm). This laser was developed specifically for this project at the Institute of High Pressure Physics, Warsaw, Poland (patent number: 9,223,123, B2; date of patent: 29 December 2015). Diode lasers are coupled to a multi-mode fibre (with a 200–400 μm core) using a reflector in the form of a regular pyramid [16]. Optimization of the optical setup makes it possible to couple 70%–90% of light into the fibre. Our laser device allows continuous wave (CW) and pulsed operation (from 1 ms upwards). All single-wavelength irradiations were performed using the same output power (1.5 W). If two wavelengths were used simultaneously, 0.75 W was set for each wavelength. In case of three wavelengths applied together, 0.5 W was set for each. For pulsed operation, a pulse width of 1 ms (10% duty cycle), 5 ms (50% duty cycle) and 8 ms (80% duty cycle) was set, with a 10 ms period (frequency 100 Hz). The 100% duty cycle corresponded to continuous wave operation. Our pulses were much longer than those of Q-switched or mode-locked lasers so we expected mainly thermal effects on the tissue. Laser spot diameter was set to 4 mm. The time of irradiation was 7 s for 100% duty cycle, 9 s for 80% duty cycle, 14 s for 50% duty cycle and 70 s for 10% duty cycle. The total dose for each irradiation was approximately 10.5 J.

### 2.2. Tissue

Irradiation was performed on the dorsal surface mucosa of a fresh (not frozen) cow’s tongue. Each irradiated area was cut off and placed in a 10% buffered formaldehyde solution. Samples were routinely processed as paraffin-embedded blocks, cut into 5 μm slices and stained with haematoxylin and eosin (HE). Each microscopic examination was performed by two independent pathologists. The following microscopic changes were examined: necrosis, occlusion of blood vessels and congestion, inflammatory reactions and obfuscation of tissue structure. Pathologists marked the border between affected tissue and normal tissue (on the digital microscopic image of the tissue). The length of the line between the border of lesions and perpendicular to the surface of the sample was a depth lesion.

### 2.3. Temperature Measurement

All irradiation procedures were recorded using a Compact Seek Thermal FLIR (forward looking infra red) Camera (Seek Thermal Inc., Santa Barbara, CA, USA). Recorded video clips were analysed using Avidemux 2.7.5 (www.avidemux.org, GNU GPL licence). Temperature was measured every 1 s. The standard deviation of maximum temperature was taken from the last three measurements. The temperature of tissue before irradiation was 27 °C. The rate of temperature increase (dT/dt) was determined as the total temperature increase divided by the total illumination time (7–70 s). However, since the increase was non-linear (faster at the beginning and slow towards the end), we also included the T(t) curves for each of the illumination parameters in Appendix A (Figure A1).

The whole experiment was repeated two times under the same conditions.

### 2.4. Statistical Analysis

Statistica version 13.3 (StatSoft, Cracow, Poland) was used to perform all statistical analyses. A statistical significance level of 0.05 was assumed. The Shapiro–Wilk test was used to check the normality of distribution. Analysis of variance (ANOVA) was applied to check temperature increase and maximum temperature differences between groups. The least significant difference post hoc test was used to check the differences in temperature between individual groups. The correlation matrix was used to estimate the correlation between lesion depth and maximum temperature and the rate of temperature increase (dT/dt). 

## 3. Results

The ambient surface temperature of tissue was 27 °C. Maximum surface temperatures after irradiation are shown in Table 1 and Figure 1. In the 10% duty cycle group, maximum temperature was observed during 450 + 520 nm (BG) irradiation; it reached 37 °C, which was significantly higher compared with other wavelengths (*p* < 0.05, Table 2). In the 50% duty cycle group, maximum temperature reached 71 °C; and in the 80% duty cycle, maximum temperature reached 89 °C, while for the 100% duty cycle, maximum temperature reached 100 °C. 

In every duty cycle group, the highest temperature was recorded during irradiation using 450 + 520 nm wavelengths. Conversely, the lowest temperature was recorded for the 640 nm wavelength; it did not exceed 39 °C in any duty cycle group. Regardless of duty cycle, the temperature reached during irradiation using the 640 nm wavelength showed statistical differences compared to other wavelengths. It is worth noting that for the 10% duty cycle group, irradiation using 520 + 640 nm wavelengths (GR) yielded a temperature of 29 °C, similar to red light. In other duty cycle groups, irradiation using 520 + 640 nm wavelengths resulted in statistical differences compared to the 640 nm wavelength. 

Combining three wavelengths during irradiation (450 + 520 + 640 nm, RGB) in the 50% and 80% duty cycle groups yielded a temperature reading statistically similar to irradiation using green light alone (520 nm). In continuous wave operation (100% duty cycle), temperature reading was statistically similar to subgroups irradiated using 520 + 640 nm wavelengths (GR) and 450 + 640 nm wavelengths (BR).

For the 50% and 80% duty cycle groups, obtained temperatures were similar between the 450 nm wavelength and 450 + 640 nm wavelengths. 

In the 100% duty cycle group, the highest temperatures, statistically comparable, were recorded for the 450 nm wavelength (B, 90 °C), 450 + 520 nm wavelengths (BG, 100 °C) and 450 + 640 nm wavelengths (BR, 86 °C). 

For continuous wavelength operation (100% duty cycle), the highest recorded temperature was 100 °C for 450 + 520 nm wavelengths (BG) and 90 °C for the 450 nm wavelength (B). No statistical differences were observed between individual groups; however, such differences did occur with respect to remaining wavelengths. The lowest temperature of 39 °C was recorded during irradiation using the 640 nm wavelength (R). 

Maximum temperatures for all studied groups are shown in Table 1 and Figure 1. A post hoc statistical analysis between the groups is shown in Table 2.

The dT/dt values for all studied groups are shown in Table 1 and Figure 2. Table 3 shows the results of post hoc analysis between the groups. The highest value of temperature increase over time (dT/dt) for the all duty cycle groups was recorded for 450 + 520 nm wavelengths (BG). In the 100% duty cycle group, the highest increase of temperature was 10.43 °C/s for 450 + 520 nm wavelengths. The lowest increase of temperature was seen during 640 nm irradiation in all of the duty cycle groups (0.03–1.57 °C/s). 

A breakdown of lesion depths is given in Table 1 and Figure 3. The results of post hoc analysis between the groups are presented in Table A1. In the 10% duty cycle group, no tissue lesions were observed under microscopic examination, regardless of the wavelength. It wass interesting that the deepest penetration of tissue lesions was observed for 450 + 520 + 640 nm wavelengths (RGB) at duty cycles of 50%, 80% and 100%. It was important that the values did not differ statistically between the 50%, 80% and 100% duty cycle groups. A similar, albeit less intense, effect (limited to only two duty cycle groups, 80% and 100%) was observed for the 450 nm (B) range. For the 520 nm (G) range and 450 + 520 nm (BG) range, the greatest depth of lesions was observed only in the 100% duty cycle group with statistical differences in relation to the other duty cycle group. The least intense lesions in tissue were observed in the 630 nm (R) range. Macroscopically, charring of the surface of the mucous membrane and, microscopically, presence of erosions were observed only in the group exposed to the 450 nm (B) and 450 + 520 nm (BG) lasers with the duty cycle coefficient of 100%. 

A positive correlation (*r* = 0.73) between the maximum obtained temperature and the depth of lesions in tissue was observed (Figure 4A). It is worth noting that, above 60 °C, the correlation coefficient was only 0.37. A positive correlation was also observed between the dT/dt value and the depth of the lesions, although in this case the correlation coefficient was 0.74 (Figure 4B).

## 4. Discussion

Despite the dynamic development of laser devices, we have not encountered in the literature research on lasers enabling simultaneous emission of three components (RGB) of visible light (red, 640 nm; green, 520 nm; blue, 450 nm) in one beam. Most of the research available in the literature compares the impact of lasers with different wavelengths on live tissue, tissue cultures or tissue preparations. However, even irradiation with different wavelengths is not conducted simultaneously, but subsequently [17,18,19,20]. Every type of laser may cause thermal damage to tissue. It can be divided into selective or non-selective. In the first case, the activity of lasers is based on the selective photothermolysis principle (ST), in which the conversion of laser energy into heat is limited to proper chromophores [21,22]. Selective destruction of target structure, with minimum thermal damage to the adjacent tissues, can be achieved through the selection of adequate wavelength (with the right absorption spectrum for a given tissue), pulse length (equal to or shorter than the time of thermal relaxation of the tissue) and energy density (enabling an effective increase of the temperature in the target tissue). The ability of tissues to dissipate heat, sometimes referred to as thermal relaxation time (TRT), is the time in which the tissue temperature is reduced to the half of beginning value [23]. The value depends on the type and volume of the tissue. If the duration of the pulse is shorter or equal to the thermal relaxation time, heat is not accumulated, which protects the adjacent structures from thermal damage. This type of action is useful when removing pigmentary lesions, vascular malformations or tattoos [24]. When selective removal of tissue is not necessary, a significant increase in temperature occurs with an increase in energy power and density and pulse width, which results in non-selective damage to tissues. Thermal effects may take the form of reversible hyperthermia when the temperature of tissues does not exceed 42 °C. Denaturation of proteins and nucleic acids occurs within the temperature range from 42 °C to 60 °C, and coagulation occurs within the range from 60 °C to 100 °C. Evaporation of water and tissue is observed above 100 °C; however, when the temperature exceeds 200 °C, tissues are charred [2]. The above phenomena are used mainly in soft tissue surgery as we can simultaneously achieve the effect of cutting, coagulation and homeostasis with the use of high-energy lasers [25]. It shortens the procedure time, improves the insight into the operating field and frequently removes the need for stitches. Most of the diode lasers available in the market enable work within a power range of up to 10 W together with regulation of pulse duration and intervals between them, the so-called Ton/Toff (Time on/Time off) [26]. Studies conducted by Matys et al. showed the impact of the operation mode of a 980 nm diode laser (SmartM, Lasotronix, Poland) on the increase in the temperature measured with the use of a TM-902-C thermometer, on a K-type probe, TP-02 (Zhangzhou Weihua Electronic Co., Fujian, China) [27]. The authors of the study observed that, regardless of the mode of operation, a significant increase in temperature occurred together with an increase in the laser power. During the pulsed mode (PM) operation, they did not observe significant differences in the increase in temperature between various pulse length settings for all ranges of the tested beam power. However, the duty cycle in all PM groups was 50% as the interval between them was proportionately extended together with the lengthening of the pulse. The aforementioned authors observed that the emission of 180 J of energy in the pulsed mode for 60 s caused an increase in temperature that was 26 °C higher compared with that of the CW mode. It may suggest that, in order to achieve a similar thermal effect, the total dose of PM energy should be decreased by approximately 30% compared to the CW dose. Our observations obtained on the basis of our own studies were contrary to this conclusion as in all groups the increase in temperature occurred together with an increase in the duty cycle value. It should be taken into account, however, that the study conducted by Matys et al. was based only on temperature measurements with the use of the K-type probe (TP-02) subjected to direct exposure to laser beam irradiation, and the measurements of the surface temperature of tissue preparations in our research were conducted with the use of a thermal imaging camera. 

Another study conducted by Fornaini et al. compared the increase in tissue temperature during the collection of specimens (ex vivo) with the use of four diode lasers with various wavelengths: 808 nm (Eufoton, Italy), 980 nm (Quanta System, Italy), 1470 nm (Quanta System, Italy) and 1950 nm (Quanta System, Italy) [28]. Power settings were 2 W and 4 W in the CW mode for all groups, with a fibre diameter of 320 µm, providing power densities of 2488 W/cm^2^ and 4976 W/cm^2^. The increase in temperature was measured with the use of two thermocouples, the first at the depth of 0.5 mm and the second at 2 mm from the incision line. The initial and final surface temperatures were recorded with the use of an IR thermometer. The greatest increase in temperature measured on the surface was observed for the 980 nm beam and 4 W (ΔT 16.3 °C), while the lowest was observed for the 1950 nm beam and 2W (ΔT 9.2 °C), which was also observed during temperature measurements in deeper tissue layers (980 nm and 4W, ΔT 40.7 °C; 1950 nm and 2W, ΔT 34.9 °C). It stems from the fact that, together with an increase in wavelength in the range from near to far infrared, the absorption of light by haemoglobin and melanin decreases (the last and the lowest of the three absorption peaks occurs at the wavelength of 920–940 nm) and the value of absorption for water increases (beginning from 700 nm, reaching the highest peak at 2500–3000 nm [29]. The authors also observed that the fastest cutting of tissue was achieved when using the 1950 beam and 4 W, and the slowest cutting was achieved when using the 880 nm beam and 2 W. It may be explained by the high value of absorption of light of 1950 nm wavelength in water. Our studies confirmed the fact that the highest tissue temperature was observed when using a beam created after combining blue and green light (BG group at all duty cycle values: 10%, 35 °C; 50%, 71 °C; 80%, 89 °C; 100%, 100 °C) as the highest absorption peaks for haemoglobin (420 nm and 520–577 nm) occurred for this wavelength. The greatest increase in temperature (dT/dt) was observed in the CW mode for wavelength BG (10.43 °C/s). 

We analysed the range of the changes that occurred in tissues and observed that the greatest depth of damage was observed during exposure to beam RGB, formed after the combining of colours R+G+B for all duty cycles except for the 10% duty cycle, and amounted to 1.43 mm (standard deviation (SD) = 0.05). In addition, no statistically significant differences in terms of their depth were observed for 50%, 80% and 100% cycle duties. The most-shallow lesions were visible for the red light (640 nm) in all duty cycle groups and ranged from 0 to 0.42 mm. Many studies presenting the scope of tissue damage as a result of the application of various laser devices can be found in the literature. Ex vivo studies, conducted by Romeo et al. with the use of lasers Er:YAG, Nd:YAG and Er-Cr:YSGG (erbium, chromium: yttrium-scandium-gallium-garnet) as well as two diode lasers (808 nm and 980 nm), showed that the 808 nm diode laser (2 W, 100 ms, duty cycle 50%, 248 J/cm^2^) and Er-Cr:YSGG (2780 nm, 3 W, 20 Hz, 53 J/cm^2^) were characterised by the smallest range of tissue damage during biopsy, which did not exceed 1 mm [30]. Moreover, comparing the 808 nm diode laser in the CW mode (2 W in power, 2400 J/cm^2^) and in the PM mode (2 W in power, 100 ms, duty cycle 50%, 248 J/cm^2^), they observed that much deeper damage occurred in the CW mode than in the PM mode (3 mm and 1 mm, respectively). 

Ex vivo studies conducted by Azevedo et al. also indicated a different range of tissue damage depth depending on the applied laser device [31]. The instruments among the studied lasers—CO_2_, 10.6 μm, 3.5 W, 50 Hz; CO_2_, 10.6 μm, 7 W, 50 Hz; CO_2_, 10.6 μm, 7 W, CW; Nd:YAG, 1.06 μm, 6 W, 40 Hz; Er:YAG, 2940 nm, 2 W, 10 Hz; laser diode, 980 nm, 3.5 W, PM)—causing the deepest damage within the margin of the collected tissue were the Nd:YAG laser and the 908 nm diode laser, and the mean depth was 670.68 μm and 626.82 μm, respectively. The most-steady cuts were observed when using the CO_2_ (3.5 W, PM) laser. It stems from the fact that the light of the 980 nm diode laser and Nd:YAG is to a lesser extent absorbed by water and to a greater extent absorbed by melanin and haemoglobin. As a result, they penetrate tissues better than the light of lasers with a larger wavelength, such as the CO_2_ laser, the activity of which is rather superficial [32].

Studies on the 445 nm diode laser (2.5 W, CW, 3100 J/cm^2^; Eltech K-Laser srl, Treviso, Italy), used during the biopsy of soft tissues within the oral cavity, showed the mean scope of thermal damage to the epithelium reaching a depth of 650.43 µm (± 311.96 µm) and damage to the subepithelial connective tissue reaching a depth of 468.23 µm (± 264.23 µm) [33]. In all samples, but one, the total thickness of thermal damage did not exceed 1 mm. Lesions of vascular aetiology, such as granuloma pyogenicum, showed much greater depth of thermal damage, which can be explained by the high absorption of light of 445 nm in wavelength for haemoglobin, occurring in large numbers in this type of lesions. Lesions diagnosed as papillomavirus showed less thermal damage due to the small share of vessels in their structure. The authors did not observe any negative impact of the laser activity on histopathological evaluation of the examined tissue. In our study, we observed a similar range of tissue penetration by a laser of 450 nm in wavelength, namely 1 mm for the 100% duty cycle. Interestingly, there were no statistically significant differences between the depth of the damage for the 80% and 100% duty cycle. The above studies suggest that most laser devices can be used for the collection of soft tissue specimens. The condition is to take into account the depths of lesions in tissues as a result of the activity of laser radiation, extending the margins of surgical excisions accordingly. A reduction of thermal damage is of key importance for histopathological evaluation. The presence of pathological tissue in the marginal area with thermal damage makes it impossible to the determine the radicality of the lesion excision and creates the risk of miscalculation of its extent, which is of key importance for further treatment. Therefore, in order to obtain a proper specimen collected with the use of a laser device, proper margin should be maintained. The margin must be greater than the depth of thermal damage characteristic for a given wavelength and laser beam parameters.

According to our research, the optimum beam for the collection of soft tissue specimens is the 450 nm beam (B). The temperature reached at 100% duty cycle (90°C) and dT/dt at the level of 9.0 °C/s are sufficient to obtain the necessary tissue cutting and coagulation effect. In addition, the depth of the thermal changes does not exceed 1 mm, thanks to which it is possible to maintain a small surgical margin during biopsy. The RGB beam (640 + 520 + 450 nm) was characterised by the greatest penetration depth (1.43 mm) among all studied groups. In addition, it caused an increase in tissue temperature (100% duty cycles, 79 °C; dT/dt, 7.43 °C) to the level enabling effective coagulation thereof. It can be applied in the treatment of shallow vascular lesions on mucous membranes, for which other laser systems are also used [34,35]. The diode lasers used most frequently in these cases are the lasers ranging from 800 nm to 980 nm as such wavelength is within one of the three absorption peaks for haemoglobin [29,36]. In addition, systems of this type rarely offer wavelengths within the range of 410–577 nm, encompassing the highest absorption peak for the protein. For this reason, the use of the RGB beam becomes promising as the components of its spectrum are within the optimum haemoglobin absorption range. Moreover, the depth of penetration was the greatest among all groups included in our study. 

## 5. Conclusions

The synergistic effect of two-wavelength (450 + 520 nm) irradiation was observed in case of maximum temperature measurement. The deepest depth of lesion was noted after three-wavelength irradiation (450 + 520 + 640 nm). Our research, although conducted ex vivo, opens new perspectives in the field of soft tissue surgery, proposing the use of diode lasers with various wavelengths in one beam. They may be useful in the selection of proper configuration when working with diode lasers in various clinical settings. However, due to the lack of reports in the literature regarding this type of laser devices (combining several wavelengths in one optic beam), it was not possible to compare our results with other studies. Thus, further research is needed in order to obtain more detailed information regarding the activity of such lasers on tissues. 

## Figures and Tables

**Figure 1 materials-13-05340-f001:**
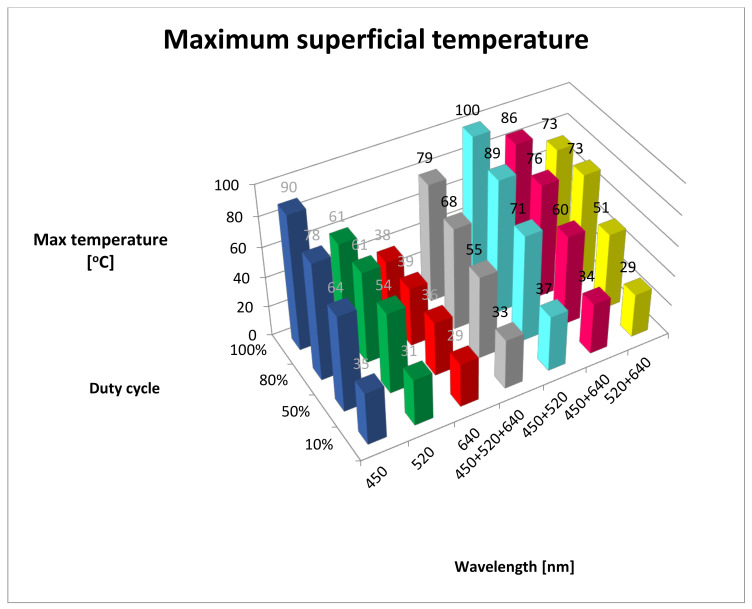
Maximum superficial temperature. B, 450 nm; G, 520 nm; R, 640 nm; RGB, 450 + 520 + 640 nm; BG, 450 + 520 nm; BR, 450 + 640 nm; GR, 520 + 640 nm).

**Figure 2 materials-13-05340-f002:**
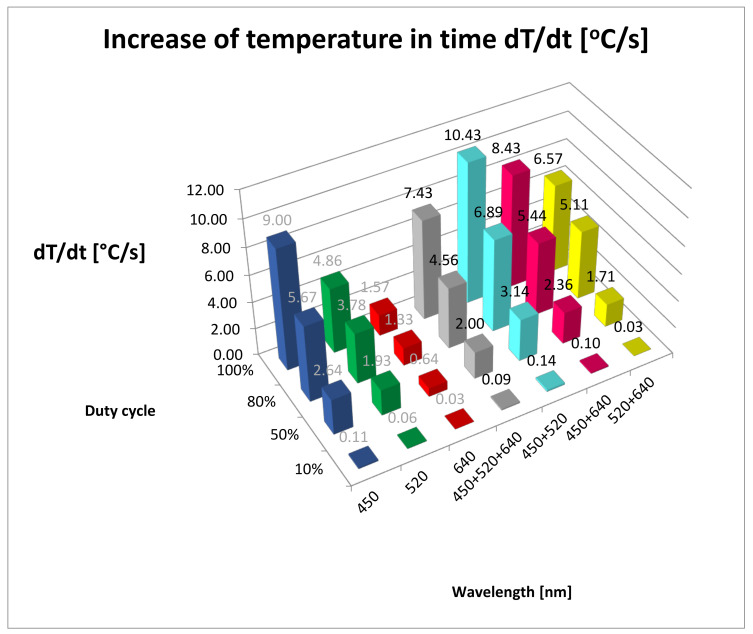
Increase of temperature in time (dT/dt (°C/s)). B, 450 nm; G, 520 nm; R, 640 nm; RGB, 450 + 520 + 640 nm; BG, 450 + 520 nm; BR, 450 + 640 nm; GR-520 + 640 nm.

**Figure 3 materials-13-05340-f003:**
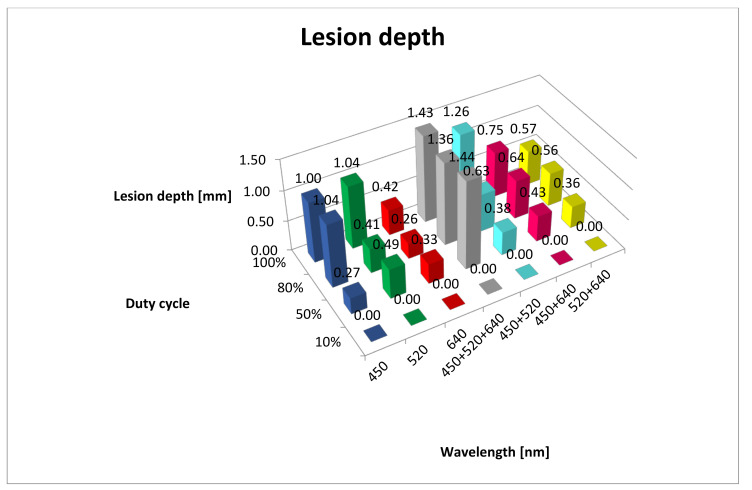
Lesion depth (mm). B, 450 nm; G, 520 nm; R, 640 nm; RGB, 450 + 520 + 640 nm; BG, 450 + 520 nm; BR, 450 + 640 nm; GR, 520 + 640 nm.

**Figure 4 materials-13-05340-f004:**
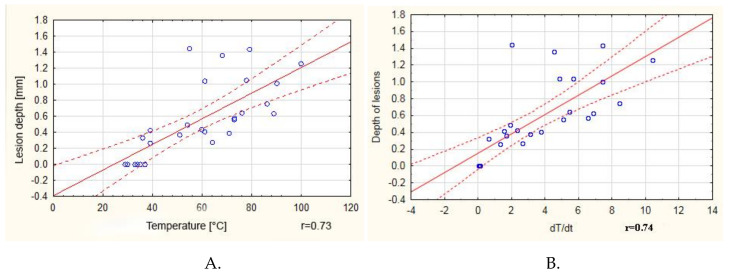
(**A**) Correlation between the maximum obtained temperature and the depth of lesions in tissue. (**B**) Correlation between lesion depth temperature and the rate of temperature increase.

**Table 1 materials-13-05340-t001:** Mean of increase of temperature in time (dT/dt), maximum temperature (Max temp) and lesion depth. SD, standard deviation; blue (B), 450 nm; green (G), 520 nm; red (R), 640 nm; RGB, 450 + 520 + 640 nm; BG, 450 + 520 nm; BR, 450 + 640 nm; GR, 520 + 640 nm; bold font, highest value in the duty cycle group.

Wavelength/ Duty Cycle (%)	dT/dt (°C/s)	SD	Max Temp (°C)	SD	Lesion Depth (mm)	SD
B10%	0.11	0.53	35.00	0.58	0.00	0.00
G10%	0.06	0.24	31.00	0.00	0.00	0.00
R10%	0.03	0.24	29.00	0.00	0.00	0.00
RGB10%	0.09	0.28	33.00	0.00	0.00	0.00
BG10%	**0.14**	0.39	**37.00**	0.00	0.00	0.00
BR10%	0.10	0.39	34.00	0.00	0.00	0.00
GR10%	0.03	0.17	29.00	0.00	0.00	0.00
B50%	2.64	2.85	64.00	1.00	0.27	0.05
G50%	1.93	2.08	54.00	0.58	0.49	0.02
R50%	0.64	0.69	36.00	0.58	0.33	0.06
RGB50%	2.00	2.15	55.00	0.58	1.44	0.02
BG50%	**3.14**	3.38	**71.00**	1.00	0.38	0.03
BR50%	2.36	2.54	60.00	1.00	0.43	0.02
GR50%	1.71	1.85	51.00	0.58	0.36	0.05
B80%	5.67	4.24	78.00	2.52	1.04	0.06
G80%	3.78	2.82	61.00	3.00	0.41	0.05
R80%	1.33	0.76	39.00	1.00	0.26	0.12
RGB80%	4.56	3.44	68.00	2.52	1.36	0.06
BG80%	**6.89**	6.69	**89.00**	2.08	0.63	0.10
BR80%	5.44	4.82	76.00	1.73	0.64	0.06
GR80%	5.11	3.06	73.00	5.51	0.56	0.07
B100%	9.00	6.66	90.00	4.04	1.00	0.24
G100%	4.86	2.66	61.00	3.51	1.04	0.07
R100%	1.57	0.41	38.00	2.00	0.42	0.02
RGB100%	7.43	6.41	79.00	6.00	1.43	0.05
BG100%	**10.43**	6.18	**100.00**	8.74	1.26	0.04
BR100%	8.43	4.62	86.00	5.51	0.75	0.04
GR100%	6.57	2.42	73.00	6.03	0.57	0.08

**Table 2 materials-13-05340-t002:** Results of post hoc statistical analysis (least significant differences) for maximum temperature. Bold font, statistically significant, *p* < 0.05; B, 450 nm; G, 520 nm; R, 640 nm; RGB, 450 + 520 + 640 nm; BG, 450 + 520 nm; BR, 450 + 640 nm; GR, 520 + 640 nm).

Maximum Temperature							
**Wavelength/** **duty cycle**	B10%	G10%	R10%	W10%	BG10%	BR10%	GR10%
B10%		**0.000000**	**0.000000**	**0.000000**	**0.000000**	**0.002189**	**0.000000**
G10%	**0.000000**		**0.000000**	**0.000000**	**0.000000**	**0.000000**	**0.000000**
R10%	**0.000000**	**0.000000**		**0.000000**	**0.000000**	**0.000000**	**1.000000**
RGB10%	**0.000000**	**0.000000**	**0.000000**		**0.000000**	**0.000064**	**0.000000**
BG10%	**0.000000**	**0.000000**	**0.000000**	**0.000000**		**0.000000**	**0.000000**
BR10%	**0.002189**	**0.000000**	**0.000000**	**0.000064**	**0.000000**		**0.000000**
GR10%	**0.000000**	**0.000000**	**1.000000**	**0.000000**	**0.000000**	**0.000000**	
Wavelength/ duty cycle	B50%	G50%	R50%	W50%	BG50%	BR50%	GR50%
B50%		**0.000000**	**0.000000**	**0.000000**	**0.000000**	**0.000022**	**0.000000**
G50%	**0.000000**		0.000000	0.141872	**0.000000**	**0.000001**	**0.000137**
R50%	**0.000000**	**0.000000**		**0.000000**	**0.000000**	**0.000000**	**0.000000**
RGB50%	**0.000000**	0.141872	**0.000000**		**0.000000**	**0.000009**	**0.000009**
BG50%	**0.000000**	**0.000000**	**0.000000**	**0.000000**		**0.000000**	**0.000000**
BR50%	**0.000022**	**0.000001**	**0.000000**	**0.000009**	**0.000000**		**0.000000**
GR50%	**0.000000**	**0.000137**	**0.000000**	**0.000009**	**0.000000**	**0.000000**	
Wavelength/ Duty cycle	B80%	G80%	R80%	W80%	BG80%	BR80%	GR80%
B80%		**0.000034**	**0.000000**	**0.000941**	**0.000192**	0.586834	0.418096
G80%	**0.000034**		**0.000000**	0.092173	**0.000000**	**0.000090**	**0.000149**
R80%	**0.000000**	**0.000000**		**0.000000**	**0.000000**	**0.000000**	**0.000000**
RGB80%	**0.000941**	0.092173	**0.000000**		**0.000000**	**0.002811**	**0.004885**
BG80%	**0.000192**	**0.000000**	**0.000000**	**0.000000**		**0.000070**	**0.000043**
BR80%	0.586834	**0.000090**	**0.000000**	**0.002811**	**0.000070**		0.784989
GR80%	0.418096	**0.000149**	**0.000000**	**0.004885**	**0.000043**	0.784989	
Wavelength/ duty cycle	B100%	G100%	R100%	W100%	BG100%	BR100%	GR100%
B100%		**0.000017**	**0.000000**	**0.010143**	0.180197	0.227531	**0.000837**
G100%	**0.000017**		**0.000269**	**0.004185**	**0.000002**	**0.000155**	**0.049241**
R100%	**0.000000**	**0.000269**		**0.000001**	**0.000000**	**0.000000**	**0.000006**
RGB100%	**0.010143**	**0.004185**	**0.000001**		**0.000628**	0.109788	0.227531
BG100%	0.180197	**0.000002**	**0.000000**	**0.000628**		**0.018204**	**0.000061**
BR100%	0.227531	**0.000155**	**0.000000**	0.109788	**0.018204**		**0.010143**
GR100%	**0.000837**	**0.049241**	**0.000006**	0.227531	**0.000061**	**0.010143**	

**Table 3 materials-13-05340-t003:** Results of post hoc statistical analysis (least significant differences) for increase of temperature. Bold font, statistically significant, *p* < 0.05; B, 450 nm; G, 520 nm; R, 640 nm; RGB, 450 + 520 + 640 nm; BG, 450 + 520 nm; BR, 450 + 640 nm; GR, 520 + 640 nm.

dT/dt
**Wavelength/** **duty cycle (%)**	B10%	G10%	R10%	RGB10%	BG10%	BR10%	GR10%
B10%		0.307940	0.126436	0.610064	0.610064	0.798710	0.123345
G10%	0.307940		0.610064	0.610064	0.126436	0.444364	0.603765
R10%	0.126436	0.610064		0.307940	**0.041767**	0.202643	0.994248
RGB10%	0.610064	0.610064	0.307940		0.307940	0.798710	0.302842
BG10%	0.610064	0.126436	**0.041767**	0.307940		0.444364	**0.040348**
BR10%	0.798710	0.444364	0.202643	0.798710	0.444364		0.198530
GR10%	0.123345	0.603765	0.994248	0.302842	**0.040348**	0.198530	
Wavelength/duty cycle (%)	B50%	G50%	R50%	RGB50%	BG50%	BR50%	GR50%
B50%		0.372804	**0.013920**	0.422296	0.532259	0.720979	0.247329
G50%	0.372804		0.110389	0.928830	0.131326	0.592311	0.788773
R50%	**0.013920**	0.110389		0.092220	**0.002316**	**0.034243**	0.182460
RGB50%	0.422296	0.928830	0.092220		0.155274	0.655344	0.720979
BG50%	0.532259	0.131326	**0.002316**	0.155274		0.327132	0.076570
BR50%	0.720979	0.592311	**0.034243**	0.655344	0.327132		0.422296
GR50%	0.247329	0.788773	0.182460	0.720979	0.076570	0.422296	
Wavelength/duty cycle (%)	B80%	G80%	R80%	RGB80%	BG80%	BR80%	GR80%
B80%		0.347996	**0.034168**	0.579935	0.542755	0.911741	0.781760
G80%	0.347996		0.225781	0.698232	0.124672	0.407219	0.506835
R80%	**0.034168**	0.225781		0.112041	**0.007317**	**0.044070**	0.063549
RGB80%	0.579935	0.698232	0.112041		0.247309	0.657763	0.781760
BG80%	0.542755	0.124672	**0.007317**	0.247309		0.472238	0.376871
BR80%	0.911741	0.407219	**0.044070**	0.657763	0.472238		0.867958
GR80%	0.781760	0.506835	0.063549	0.781760	0.376871	0.867958	
Wavelength/duty cycle (%)	B100%	G100%	R100%	RGB100%	BG100%	BR100%	GR100%
B100%		0.161044	**0.014214**	0.591242	0.625293	0.844939	0.407680
G100%	0.161044		0.264239	0.380892	0.061831	0.225558	0.558100
R100%	**0.014214**	0.264239		0.050103	**0.003950**	**0.022914**	0.092439
RGB100%	0.591242	0.380892	0.050103		0.307440	0.732273	0.769303
BG100%	0.625293	0.061831	**0.003950**	0.307440		0.494752	0.191235
BR100%	0.844939	0.225558	**0.022914**	0.732273	0.494752		0.525921
GR100%	0.407680	0.558100	0.092439	0.769303	0.191235	0.525921

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
