# Peer review of "Assessment of Effects of Laser Light Combining Three Wavelengths (450, 520 and 640 nm) on Temperature Increase and Depth of Tissue Lesions in an Ex Vivo Study"

_materials, 2020, doi:10.3390/ma13235340_

Round 1

Reviewer 1 Report

The paper entitled "Assessment of effects of laser light combining three wavelengths (450, 520 and 640 nm) on temperature increase and depth of tissue lesions in ex-vivo study" is an original paper to assess the effects of laser radiation in the visible spectrum on tissue in vitro, according to wavelength and pulse width.

The whole text is well written both in syntax and English grammar. 

Introduction is exhaustive and report the right and recent pertinent references. 

M&M are adequately reported to reproduce the experiments. What it is not clear, is how many times the experiment was reproduced. I suggest authors to report it. 

Results are well supported by graphs and tables. 

Table 2 and 3 reported the statistically significant values in red. I suggest to prefer a grayscale (or use bold type), to allow any reader, even those with vision impairments, to grasp the results.

discussion: is consistent and well argued.

Please, correct "Romero et al " in Romeo, line 273 

conclusion: are well written and explained the future perspectives. 

Author Response

Dear Reviewer,

Thank you for your suggestions. In table 2 and 3 we changed red color into bold font for significant differences. Romeo et al. was corrected in 273 line. We added chart of increase temperature in time as a supplementary material.

Best regards,

Authors.          

Reviewer 2 Report

The authors have investigated the effect of laser light, combining three wavelengths (450, 520 and 640 nm) on tissue  and studied e.g. the increase of temperature and depth of tissue lesions by the relatively intense (up to 1.5W laser power) radiation.

In their introduction they are explaining the reasons for using pulsed laser excitation and that they mix up to three wavelengths, while keeping the nominal CW equivalent power constant. 

There are couple of statement which are a bit disturbing.

First, they state: "Thanks to monochromaticity, laser beams can selectively affect chromophores found in tissues, as they absorb electromagnetic waves with predilection to their specific length. There are four main tissue
40 chromophores, each absorbing specific wavelengths: water (1400 – 10000 nm), haemoglobin (410 – 600 nm and 800 – 1100 nm), hydroxyapatite (2800 nm) and melanin (400 – 800 nm) [5,9]."

The given wavelengths ranges are huge covering e.g. the full visible range or a larger part of the NIR Spectrum. Monochromaticity is surely not required here - at least not for this reason.

BTW, the high power diode laser used will be fairly narrow in their spectral features but not strictly monochromatic anyway - at least not in the physics sense of it.

The laser is decribed to be pulsed. However, again not in the physics sense of the word. Pulsed lasers are able to create new effects which can be chemical in nature or even ablation which requires that the removal of tissue is faster than the thermal conductivity and hence the matter is removed without damaging the surrounding tissue. This is highly of interest and therefore used e.g in cutting processes e.g. in eye surgery.

The timescale for such processes reach from ns to femtoseconds as these short timescale are capable of delivering huge powers to the sample.

However, here, the author merely use what we would call a modulated laser with a duty cycle between 10% and 100% (CW) and pulse lengths between 1 ms and 10 ms. In these time domains the "pulsing" doesn't change much as the tissue is burned and thermally destroyed.

Also using three different but fixed wavlengths is not really that helpful. Absorption will vary dependnig on the absorbance (which is directly connected to its colour) of the tissue.

Furthermore, the authors talk about that the three colors "create white light".

However, this is just an illusion as the human eye has only three distinct kinds of colour receptors and thus doesn't perceive the wide spectral gaps, but is definitely not an argument  in favour of the article. The tissue frankly doesn't care about our eyes.

This is then reflected in the results which do not show strong effects depending on the used radiation. It would have been more interesting to use a tunable laser and to use dedicated wavelenghts which are only absorbed by certain chemical groups, organelles, dyes etc....

However, the above statements must be corrected and explained more carefully.

Furthermore, I suggest to cite the following publication as a very good example of a laser induced photodynamic therapy which is analysed by a number of complementary techniques (even at the nanoscale):

"AFM-based bivariate morphological discrimination of apoptosis induced by photodynamic therapy using photosensitizer-functionalized gold nanoparticles"

R Al-Majmaie, et. al. RSC Advances 5 (101), 82983-82991 (2015)

Author Response

Dear Reviewer,

We added following changes according to your suggestions:

First, they state: "Thanks to monochromaticity, laser beams can selectively affect chromophores found in tissues, as they absorb electromagnetic waves with predilection to their specific length. There are four main tissue
40 chromophores, each absorbing specific wavelengths: water (1400 – 10000 nm), haemoglobin (410 – 600 nm and 800 – 1100 nm), hydroxyapatite (2800 nm) and melanin (400 – 800 nm) [5,9]."

The given wavelengths ranges are huge covering e.g. the full visible range or a larger part of the NIR Spectrum. Monochromaticity is surely not required here - at least not for this reason.

BTW, the high power diode laser used will be fairly narrow in their spectral features but not strictly monochromatic anyway - at least not in the physics sense of it.

We added (approximate) monochromaticity.

We agree that such wide absorption ranges are meaningless. We changed the above sentence to: In the visible range (400-700 nm) the absorption in the tissue is dominated by hemoglobin (with peaks in the blue and yellow region) and by melanin (decreasing monotonically towards longer wavelengths). With our three wavelengths we expect the strongest absorption for the blue, intermediate for the green and lowest for the red. The penetration depth should increase for decreasing absorption i.e. highest for the red, intermediate for the green and lowest for the blue.

The laser is decribed to be pulsed. However, again not in the physics sense of the word. Pulsed lasers are able to create new effects which can be chemical in nature or even ablation which requires that the removal of tissue is faster than the thermal conductivity and hence the matter is removed without damaging the surrounding tissue. This is highly of interest and therefore used e.g in cutting processes e.g. in eye surgery.

The timescale for such processes reach from ns to femtoseconds as these short timescale are capable of delivering huge powers to the sample.

However, here, the author merely use what we would call a modulated laser with a duty cycle between 10% and 100% (CW) and pulse lengths between 1 ms and 10 ms. In these time domains the "pulsing" doesn't change much as the tissue is burned and thermally destroyed.

We agree that our pulses are much longer than for Q-switched or mode-locked lasers but still these are pulses of varying duration (generated by pulsed current supply). We added a sentence: Our pulses are much longer than for Q-switched or mode-locked lasers so we expect mainly thermal effects on the tissue. 

Also using three different but fixed wavlengths is not really that helpful. Absorption will vary dependnig on the absorbance (which is directly connected to its colour) of the tissue.

Furthermore, the authors talk about that the three colors "create white light".

However, this is just an illusion as the human eye has only three distinct kinds of colour receptors and thus doesn't perceive the wide spectral gaps, but is definitely not an argument  in favour of the article. The tissue frankly doesn't care about our eyes.

We changed “white light” into “three components of visible light”

This is then reflected in the results which do not show strong effects depending on the used radiation. It would have been more interesting to use a tunable laser and to use dedicated wavelenghts which are only absorbed by certain chemical groups, organelles, dyes etc....

We agree that a study with a tunable laser (like Titanium-Sapphire or dye laser) would be much more complete than our “three-wavelength” study. However, our main purpose was to study the possible synergies of using two or three wavelengths simultaneously. And the result that Blue+Green illumination heats the tissue more effectively than individual colors or other color combinations was rather unexpected.

However, the above statements must be corrected and explained more carefully.

Furthermore, I suggest to cite the following publication as a very good example of a laser induced photodynamic therapy which is analysed by a number of complementary techniques (even at the nanoscale):

We added  in the introduction one more reference: "AFM-based bivariate morphological discrimination of apoptosis induced by photodynamic therapy using photosensitizer-functionalized gold nanoparticles" R Al-Majmaie, et. al. RSC Advances 5 (101), 82983-82991 (2015).

Whole manuscript was checked by native speaker.

Best regards,

Authors.

Round 2

Reviewer 2 Report

Some points have been clarified and some additions have been made.

There is still room for improvement but any futher changes would be time consuming and hence maybe asking for too much.

Therefore, I believe the paper can now be accepted as is.